# Confirmation of the topology of the Wendelstein 7-X magnetic field to better than 1:100,000

T. Sunn Pedersen[1,2], M. Otte[1], S. Lazerson[3], P. Helander[1,2], S. Bozhenkov[1], C. Biedermann[1], T. Klinger[1,2], R.C. Wolf[1,4], H.-S. Bosch[1,4] & The Wendelstein 7-X Team[†]

Fusion energy research has in the past 40 years focused primarily on the tokamak concept, but recent advances in plasma theory and computational power have led to renewed interest in stellarators. The largest and most sophisticated stellarator in the world, Wendelstein 7-X (W7-X), has just started operation, with the aim to show that the earlier weaknesses of this concept have been addressed successfully, and that the intrinsic advantages of the concept persist, also at plasma parameters approaching those of a future fusion power plant. Here we show the first physics results, obtained before plasma operation: that the carefully tailored topology of nested magnetic surfaces needed for good confinement is realized, and that the measured deviations are smaller than one part in 100,000. This is a significant step forward in stellarator research, since it shows that the complicated and delicate magnetic topology can be created and verified with the required accuracy.

[1] Max Planck Institute for Plasma Physics, Wendelsteinstrasse 1, 17491 Greifswald, Germany. [2] University of Greifswald, Domstrasse 11, 17489 Greifswald, Germany. [3] Princeton Plasma Physics Laboratory, PO Box 451, Princeton, New Jersey 08543, USA. [4] Technical University Berlin, Strasse des 17. Juni 135, 10623 Berlin, Germany. Correspondence and requests for materials should be addressed to T.S.P. (email: thomas.sunn.pedersen@ipp.mpg.de).
[†] A full list of consortium members appears at the end of the paper.

Fusion has the potential to cover the energy needs of the world's population into the distant future. Lawson showed in 1957 that magnetic confinement fusion based on deuterium–tritium fusion can work as a net energy source if one achieves a sufficiently high triple product, $n_i T_i \tau_E > 4 \times 10^{21}\,\mathrm{keV\,m^{-3}\,s}$ for the plasma, approximately valid for ion temperatures $T_i$ in the range $10$–$40\,\mathrm{keV}$ (ref. 1).

Here $n_i$ is the ion density, and $\tau_E$ is the energy confinement time, which for a typical operating point in magnetic fusion reactor studies is a few seconds.

A promising approach to meeting this challenge is the use of a magnetic field that creates toroidal magnetic surfaces.

Of these concepts, the tokamak has so far shown the best confinement properties, but the stellarator is not far behind, and there is reason to believe that it can catch up. In a stellarator, nested toroidal magnetic surfaces are created from external magnetic coils, see Fig. 1. Each magnetic field line meanders around on its magnetic surface; it never leaves it. In general, if one follows a field line from one point on a magnetic surface, one never comes back to the same exact location. Instead, one covers the surface, coming infinitely close to any point of the surface.

The stellarator is different from the other toroidal magnetic surface concepts in that both the toroidal and the poloidal field components—which together create the magnetic surface topology—are created from currents in external coils. In the tokamak and the reversed-field pinch[2], a strong toroidal current driven within the plasma is needed to generate the poloidal magnetic-field component. The stellarator's lack of a strong current parallel to the magnetic field greatly reduces macroscopic plasma instabilities, and it eliminates the need for steady-state current drive. This makes it a more stable configuration, capable of steady-state operation. These are important advantages for a power plant.

The stellarator was invented by Lyman Spitzer in the 1950s (ref. 3). So why did it fall behind? And why do some believe that it is about to have a comeback?

Plasma confinement in early stellarators was disappointing. This was due to poorly confined particle orbits—many of the particle trajectories were not fully confined, even though the magnetic field lines were. If each guiding centre (the point around which the particle performs its rapid gyration) were to stay exactly on the magnetic field line it starts out on, the magnetic surfaces would guarantee good confinement. But for all toroidal magnetic systems, the orbits deviate from the field lines, since the guiding centres drift perpendicular to the magnetic field. This is due to the field-line curvature and magnetic field strength inhomogeneities inherent to the toroidal magnetic topology. In a magnetically confined fusion plasma, the drift is on the order of 10,000 times slower than the particle velocity, but, at $100\,\mathrm{m\,s^{-1}}$, it will lead to particle losses in less than 1/10 of a second, if the drifts do not average out or stay within the magnetic surface, but instead carry the particle from the inner to the outer magnetic surfaces. This was the case in early stellarator experiments. The tokamak and the reversed-field pinch do not suffer from this problem since their toroidal symmetry makes the particle drifts average out for all the particles and therefore only cause minor excursions from the magnetic surface.

Advances in plasma theory, in particular in the 1980s and 1990s, allowed the development of stellarator magnetic field configurations that display greatly improved confinement (see refs 4,5), reducing the drift orbit losses to a level sufficiently small so that it is predicted to be compatible with an economically feasible fusion power plant. The optimization itself, as well as the associated design of coils that realize the optimized magnetic fields, requires computer power that only became available in the 1980s. The first generation of optimized stellarators started operation in the 1990s, and confirmed many of the expected improvements[6,7]. These devices were, however, too small to reach the high ion temperatures where the optimization really comes to its test. Moreover, they were built with copper coils, which are adequate for proof-of-principle studies but incompatible with steady-state operation at high magnetic field strengths. The Wendelstein 7-X (W7-X) stellarator experiment is the first representative of the new generation of optimized stellarators, and aims to show with its superconducting coil system and relatively large size (major radius 5.5 m), quasi-steady state operation with plasma parameters, including ion temperatures, close to those of a future fusion power plant[8,9]. The sophisticated computer optimization of W7-X came at a price, however: the coils have complicated three-dimensional (3D) shapes, reminiscent of sculptures, Fig. 1. With today's 3D design and manufacturing techniques, complex 3D engineering has become feasible, albeit still challenging[10]. Strict requirements for the manufacturing and assembly accuracy of the coils add to the engineering challenge, which was in fact viewed by some as unrealistic. High engineering accuracy is needed because small magnetic field errors can have a large effect on the magnetic surfaces and the confinement of the plasma.

The measurements that are presented in the following sections confirm that the engineering challenges of building and assembling the device, in particular its coils, with the required accuracy, are met successfully. To explain how this was done, we first describe a few key concepts.

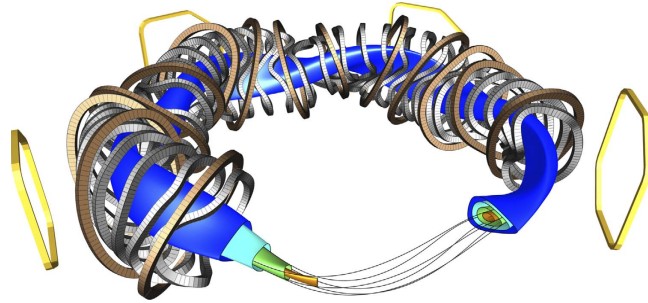

**Figure 1 | Layout of W7-X.** Some representative nested magnetic surfaces are shown in different colours in this computer-aided design (CAD) rendering, together with a magnetic field line that lies on the green surface. The coil sets that create the magnetic surfaces are also shown, planar coils in brown, non-planar coils in grey. Some coils are left out of the rendering, allowing for a view of the nested surfaces (left) and a Poincaré section of the shown surfaces (right). Four out of the five external trim coils are shown in yellow. The fifth coil, which is not shown, would appear at the front of the rendering.

## Results

**Hamiltonians and magnetic surfaces.** The equations governing magnetic field lines can be written in Hamiltonian form. It is curious that this simple, but little-known, fact was discovered only half a century ago[11], but thanks to it, the entire arsenal of Hamiltonian chaos theory can be applied to magnetic fields. For instance, the celebrated Kolmogorov–Arnold–Moser (KAM) theorem[12–14] guarantees that small perturbations to an otherwise integrable magnetic field preserve the topology of most field lines, and break it by generating so-called magnetic islands only at well-defined locations. As we shall see, these islands can be measured and visualized directly in W7-X and offer the opportunity to detect field perturbations smaller than $\delta B/B \sim 10^{-5}$. To our knowledge, it is the first time that the topology of a magnetic field has been measured so accurately. For more information on the

theory of shaped magnetic fields and their role in plasma confinement, we refer to two recent reviews[15,16].

A magnetic surface is not only characterized by its shape and enclosed volume, but also by its rotational transform, $\iota$. This is a measure of the poloidal rotation ('twist') of the field lines as one follows them around the magnetic surface; $\iota = 1/2$ indicates that the field line moves halfway around a magnetic surfaces in the poloidal direction for each toroidal turn it makes. Thus, for $\iota = 1/2$, the field line bites itself in the tail after two toroidal transits. Since there are many more irrational than rational numbers, $\iota$ is typically irrational, and a magnetic field line generally does not close on itself, it densely traces out a two-dimensional surface.

**Measuring the magnetic topology.** Since the magnetic surface topology in a stellarator is created entirely from external coils, it can be measured in the absence of a plasma. This is done using an electron beam injected along the magnetic field. It follows and therefore maps out the magnetic field lines, and thus allows confirmation of the magnetic surface topology, providing a flux surface map. As mentioned earlier, the motion along the field is much faster than the guiding-centre drifts. This is even more so for the relatively low-energy electrons used in magnetic-surface mapping. Owing to the launch of the electrons parallel to the magnetic field, and the much smaller mass of electrons relative to any ion, its ratio of parallel velocity to guiding centre drift velocity is of order 1 million. Thus, the beam follows the magnetic field lines to a very high accuracy. The source of the electron beam is an electron gun, a small negatively biased and heated thermionic electron emitter surrounded by a small electrically grounded cylindrical structure. This electron beam alone can visualize the magnetic field line on which it is placed, through collisional excitation of a dilute background gas inside the vacuum chamber. This way, striking images can be made of the 3D structure of the magnetic surfaces; see Fig. 2 and refs 17,18.

A two-dimensional cross-sectional image generally provides clearer information though, just as Poincaré phase-space maps do for other Hamiltonian systems. Such a Poincaré plot of the magnetic surface is realized experimentally by intersecting the electron beam with a rod covered with a fluorescent, here a special zinc oxide powder (ZnO:Zn). When the rod intersects the magnetic surface on which the electron beam circulates,

it fluoresces at the one or usually two locations where the rod intersects the magnetic surface and therefore collides with the electron beam. As the rod moves through the surface, all points on the latter will eventually light up. In a long camera exposure of this sweep motion, the entire cross-section of the magnetic surface appears, as shown in Fig. 3. The motion of the rod itself is often invisible on such an image, since the light sources (other than the fluorescence) are kept as weak as possible. After an exposure, one can move the electron gun to another field line that defines another magnetic surface, and repeat the process. This way, the nested, closed magnetic surface topology, which is illustrated in Fig. 1, can be experimentally verified[19–22], and if any magnetic island chains exist, they will show up in the Poincaré plot, as explained in the following.

**Island chains and error fields.** An island chain can appear on any magnetic surface with a rational value of $\iota$: a direct confirmation of the small-denominator problem in KAM theory[12]. In practice, island chains with a detectable and operation-relevant size only appear for low-order rational values of $\iota$, and only if there is a Fourier component of the magnetic field that has matching (that is, resonant) toroidal and poloidal mode numbers, $n$ and $m$, so that $\iota = n/m$.

W7-X is designed to reach $\iota = 1$ at the outermost flux surface. It is a fivefold periodic device, with a pentagon-like shape, and thus has an $n = 5$ Fourier component to its magnetic field, so that an $n = m = 5$ island chain appears at the plasma edge. We denote unwanted field components error fields, and describe them in relative terms, $b_{mn} = B_{mn}/B_0$, where $B_0$ is the average magnetic field strength in the confinement region, and $B_{mn}$ is the amplitude of the Fourier component of the error field. In the search for error fields, we focus on the toroidal $n$ numbers since only $n = 5$ and multiples thereof should be present, whereas a broad spectrum of poloidal $m$ numbers is present in W7-X. The $n = 1$ through 4 components are to be avoided as much as possible, to ensure symmetric heat load distributions onto the $2 \times 5 = 10$ divertor units to be installed at the vessel wall in future operation phases[23]. For the symmetry-breaking $n = 1$ through 4 error fields, deformations due to electromagnetic forces do not play a major role and the $b_{mn}$'s are largely independent of the magnitude of $B_0$, in contrast to the effects discussed in the 'Discussion' section. Of particular concern is the $n = 1$ component, which would create an $n/m = 1/1$ island chain, and would result from, for example, a slightly misplaced coil module.

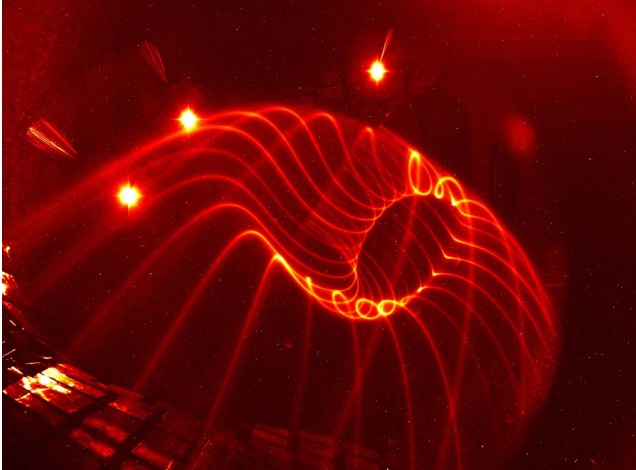

**Figure 2 | Experimental visualization of the field line on a magnetic surface.** The field lines making up a magnetic surface are visualized in a dilute neutral gas, in this case primarily water vapour and nitrogen ($p_n \approx 10^{-6}$ mbar). The three bright light spots are overexposed point-like light sources used to calibrate the camera viewing geometry.

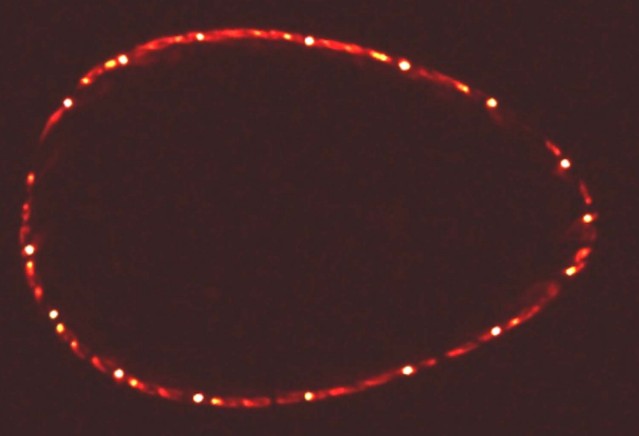

**Figure 3 | Poincaré section of a magnetic surface.** The Poincaré section of a closed magnetic surface is measured using the fluorescent rod technique. The electron beam circulates more than 40 times, that is, over 1 km along the field line.

When minimizing the error fields, the main engineering challenge is the geometrical precision during coil manufacturing and coil assembly. The $3.5 \times 2.5 \times 1.5$ m-size non-planar coil winding packs with their five different geometries (cf. Fig. 1) are particularly critical[24]. The construction of W7-X required, for the first time, industry to manufacture superconducting coils with a highly complex shapes, with tolerances in the $\pm 1$ mm regime. This was accomplished by using specialized winding devices combined with precision metrology[25].

It was even more challenging to maintain the precision, and keep track of it, during installation of the coils: Positioning of the coils, machining of the contact elements, welding of mechanical supports and bolting to the massive central support ring, all sums up to create an additional contribution to the error field. It was only possible to keep deviations during installation and assembly into coil groups under control by intensive use of laser-based metrology tools, systematic adjustment procedures, as well as advanced welding and machining technologies. The largest coil placement errors were less than 4.4 mm, resulting in an expected largest Fourier coefficient of the magnetic perturbation error of $b_{11} \approx 1.2 \times 10^{-4}$ (ref. 26).

**Measuring error fields.** Magnetic flux surface mapping, in particular of island chains[27], allows for detailed error field detection and correction[19,28]. Island chains are sensitive indicators of small changes in the magnetic field topology, since they are physical manifestations of resonances in the magnetic topology. The radial full width $w$ of an island chain is related to a resonant magnetic field component through ref. 16

$$w = 4\sqrt{\frac{R_0 B_{mn}}{m(d\iota/dr)}} \Leftrightarrow B_{mn} = \frac{d\iota}{dr}\frac{w^2 m}{16 R_0}. \quad (1)$$

The width of an island chain depends on the square root of the resonant field component, $B_{mn}$, with $\iota = n/m$, and the magnetic shear $d\iota/dr$, as well as the poloidal mode number $m$ and the size of the device (via the major radius $R_0 = 5.5$ m in W7-X). In W7-X, the rotational transfrom $\iota$ is nearly constant from the inner to the outer magnetic surfaces, then $d\iota/dr$ is small, and a sizeable island chain will result from even a very small resonant error field.

With field-line mapping, island chains can be detected, and, thus, $\iota$ can be determined at a specific radial location, and resonant error fields, if present, can be measured.

We show in the following that effects due to slight deformations of the magnetic coils are clearly visible, and that an important error field component in W7-X has been measured to be less than 1 in 100,000. To our knowledge, this is an unprecedented accuracy, both in terms of the as-built engineering of a fusion device, as well as in the measurement of magnetic topology.

**Adjustment of $\iota$.** The magnetic topology used for initial plasma experiments in W7-X was chosen so as to avoid island chains at the plasma edge[29].

The rotational transform $\iota$ varies from 0.79 in the centre to 0.87 at the outer magnetic surface that just touches the graphite

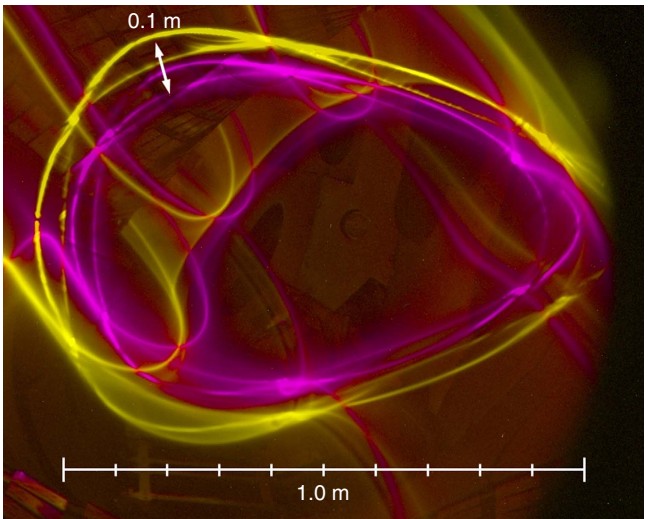

**Figure 5 | Island chain shifts at higher field.** The 5/6 island chain is shown in cyan for $B = 0.4$ T, and in yellow for $B = 2.5$ T. Although nominally one might expect them to be identical, the 5/6 island chain is about 10 cm further out at high field strength, due to small deformations of the magnet coils under electromagnetic forces.

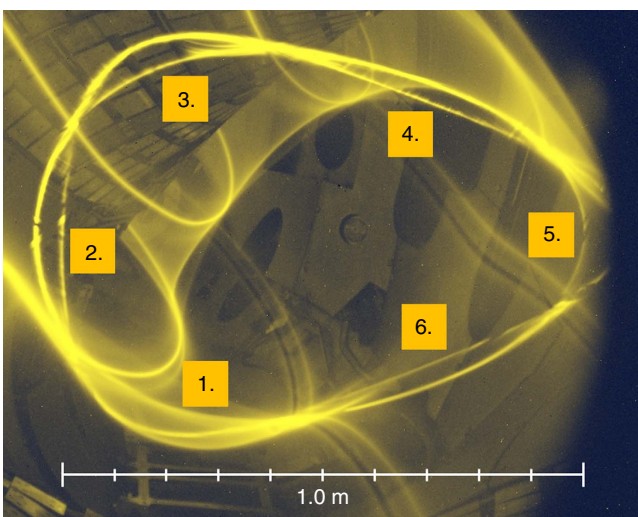

**Figure 4 | The natural 5/6 island chain.** The 5/6 island chain is visible in a poloidal-radial Poincaré plot created by an electron gun and a sweep rod, as a set of six 'bubbles', reflecting the $m = 6$ poloidal mode number. A thin background gas in the chamber creates a visualization of the field lines that create the x-points of the island chain.

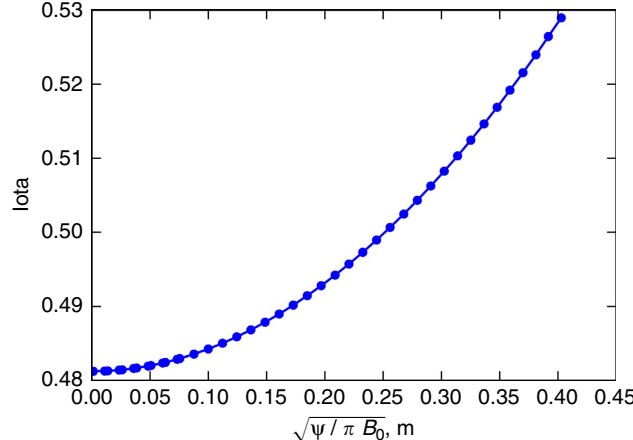

**Figure 6 | Profile of $\iota$ for error field studies.** The $\iota$ profile is shown for the special configuration developed for field error detection. The $\iota$ varies only minimally around the resonant value of $1/2$. The x axis is a measure of the minor radial size (in meters) of the magnetic flux surface, that is, a pseudo-radial coordinate.

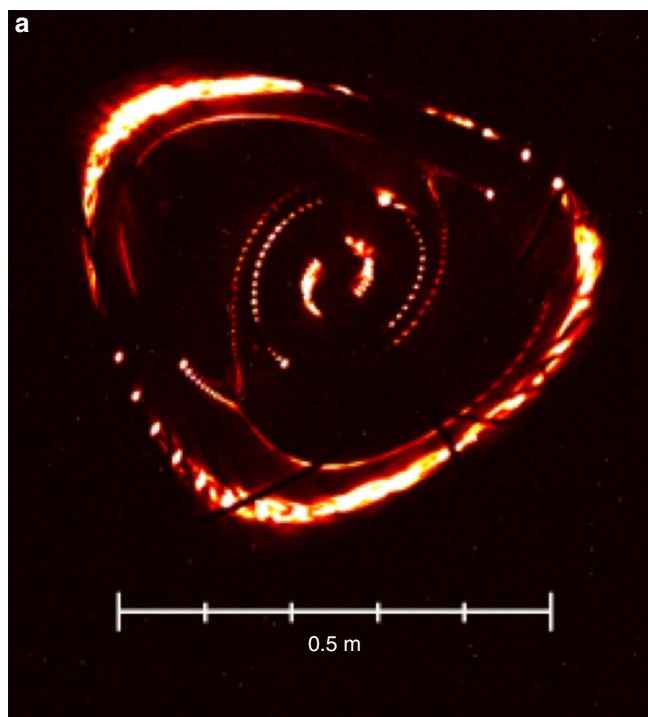

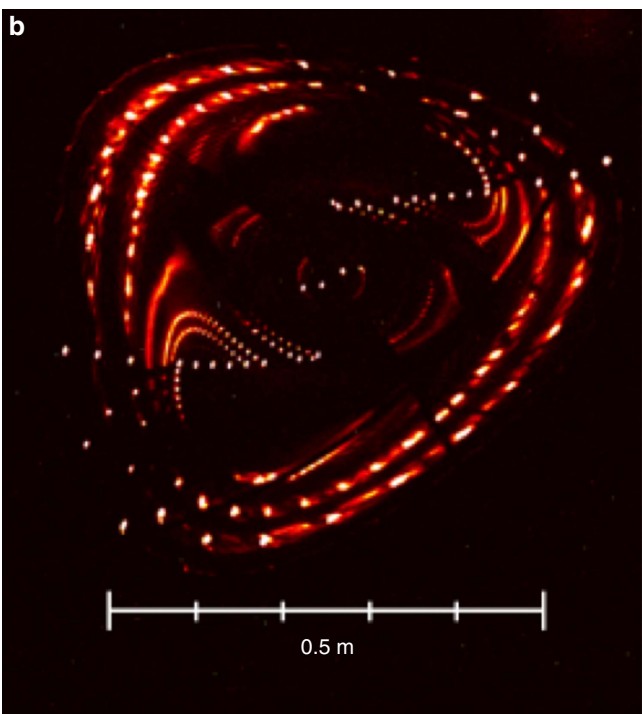

**Figure 7 | Measured island chains for different coil current settings.** For the special $\iota \approx 1/2$ configuration, the $n=1$, $m=2$ island size and phase can be measured by the Poincaré section method. Here two conglomerate images **a** and **b** with several nested surfaces are shown for two different phases of a purposely added $n=1$ field structure with the same amplitude. Although the shadowing problem leads to gaps, the trained eye can still detect the changes in size and phase of the $m=2$ island.

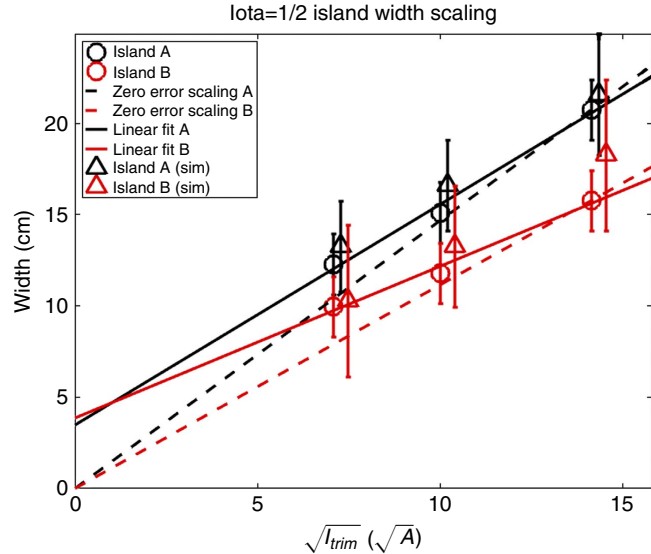

**Figure 8 | Comparison with metrology-generated numerical model.** The measured island widths are compared directly with those predicted from numerical calculations that take the as-built as-installed geometry of the W7-X coil set into account. Excellent agreement is seen. The offset from zero in the linear fits indicate the intrinsic 4 cm island width. If no intrinsic error field were present, the points would have lined up with the dotted lines. The island widths are determined from the real or synthetic images by use of an image processing software programme developed for these purposes. Since it was not always possible to image the edge of the island chain exactly, and even when so, the electron beam gives a certain width to an island chain or a magnetic surface, the island width has some uncertainty. The error bars indicate the largest and smallest possible island size consistent with the data.

limiters installed to protect in-vessel components by intercepting the plasma heat loads.

The $\iota = 5/6 \approx 0.83$ resonance is located inside the confinement region—and is thus unproblematic for the plasma-facing components. It creates a prominent island chain, because of the built-in $n=5$ component in W7-X. This island chain is indeed clearly visible, as seen in Fig. 4 showing a measurement performed at the field strength $B = 2.5\,\mathrm{T}$ later intended for plasma operation. The island chain location was detected almost exactly at the position expected from calculations taking the elastic deformation of the superconducting coils into account. These deformations, due to the electromagnetic forces between the magnets, cause a roughly 1% decrease in $\iota$, thus shifting the location of $\iota = 5/6$ a few centimetres outward from where they would be without coil deformation. This was confirmed by repeating the measurements at $= 0.4\,\mathrm{T}$ and observing that the island chain indeed appears those few centimetres further inward, Fig. 5. At $B = 0.4\,\mathrm{T}$, the electromagnetic forces are $(2.5/0.4)^2 \approx 39$ times smaller than at $B = 2.5\,\mathrm{T}$. The actual change in the angle of the magnetic field vector detected in this way is only about 0.1%. Nevertheless, it shows up in Fig. 5 as a clearly visible radial shift of the island chain. A more detailed analysis of these data can be found elsewhere[30].

**Evaluation of an important error field component.** For the first measurements of the $n=1$ error field, a special magnetic surface configuration was used[31], where $\iota$ varies slowly and passes through the resonance $\iota = 1/2$, see Fig. 6.

In the complete absence of error fields, a small $n=5$, $m=10$ island chain would appear at the $\iota = 1/2$ location at around 25 cm distance from the innermost magnetic surface, but in the presence of even a small $n=1$ error field, an $n=1$, $m=2$ island chain, visible in a Poincaré plot as two 'bubbles', will appear.

The $B_{21}$ error field is too small to create an island structure large enough to be measured clearly. This is in part due to the good news that it is small, and in part due to $\iota$ being so close to 1/2, that the electron beam comes very close to its launch position

(the electron gun) after two toroidal transits, thus running the risk of hitting the back of the electron gun and disappearing.

It is nevertheless possible to indirectly measure the $B_{21}$ field error, despite this shadowing problem, by adding an $n=1$ error field with a well-defined amplitude and phase, using the set of five large external coils[32], four of which are shown in yellow in Fig. 1. The primary purpose of these coils is to trim away the unwanted $n=1$ error field components, but the trim coils are used here to create an extra $n=1$ error field, and thus generate an $n/m=1/2$ island chain wide enough to be measurable.

Light fibres installed in the vessel along with detailed measurements of their location allow the pixels of the image plane to be mapped to physical dimensions. In this way, the width of the island in physical units can be inferred from a measurement in pixels. Error bars account for both the physical width of the flux surface traces and the step size going from outside the island chain to inside it. A best attempt is made to report the maximum width of the magnetic islands.

By scanning the phase and amplitude of the imposed, well-defined error field, measuring the island phase and width (Fig. 7), and comparing with equation 1, we find that an $n/m=1/2$ island with a width of 4 cm must be present, even in the absence of trim-coil induced fields.

The configuration has $d\iota/dr\approx0.15\,\text{m}^{-1}$ at the $\iota=1/2$ location, so using equation (1) again, we arrive at $B_{21}\approx5.4\times10^{-6}$. This value is well within the range that can be corrected with the trim coils[32]. The careful and accurate metrology described earlier in this article is validated by our flux-surface measurements: The as-built coil forms and their as-installed locations have been implemented numerically in our codes, and then used to calculate the size, phase and location of the intrinsic 1/2 island chain resulting from the $B_{21}$ component. These data agree very well with our fully independent direct measurements of the magnetic topology. The agreement regarding amplitude is shown in Fig. 8. Good agreement is obtained not only for the amplitude of the island chain but also its phase.

## Discussion

The now experimentally validated numerical model of the coil system allows us to identify the primary source of the measured error field. The measured field error is caused primarily by imperfections in the placement and shapes of the planar coils. For the special magnetic configuration chosen here, the planar coils produce a much larger fraction of the magnetic field than they do in configurations used for plasma operation; in fact the one major configuration that has $\iota=1$ at the plasma edge has no planar coil current. Therefore, we plan to measure the $B_{11}$ error field in a configuration whose magnetic field is overwhelmingly dominated by the non-planar coils with $\iota\approx1$ (ref. 33). Since the $B_{11}$ and the $B_{21}$ components should be roughly of the same order of magnitude, and since the $B_{21}$ error is reproduced by our numerical models, the $b_{11}$ error is also expected to be small, likely close to or somewhat below the aforementioned estimate of $1.1\times10^{-4}$, thus well within the correction capabilities of the W7-X coil set.

The need for complex 3D shaping and high-accuracy requirements have been viewed as major problems for optimized stellarators. Wendelstein 7-X demonstrates that a large, optimized, superconducting stellarator can be built with an accuracy sufficient to generate good magnetic surfaces with the required topology, and that experimental tools exist to verify the magnetic topology down to and below errors as small as 1:100,000. These results were obtained using magnetic field-line mapping, a sensitive technique to measure the detailed topology of the magnetic surfaces. To reach the other goals of the device, and provide an answer to the question 'is the stellarator the right concept for fusion energy?', years of plasma physics research is needed. That task has just started.

**Data availability**. The data sets generated and/or analysed during the current study are available from the corresponding author on reasonable request.

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

## Acknowledgements

This work has been carried out within the framework of the EUROfusion Consortium and has received funding from the Euratom research and training programme 2014–2018 under grant agreement No 633053. The views and opinions expressed herein do not necessarily reflect those of the European Commission. S.L. acknowledges support from US DOE grant DE-AC02-09CH11466. T.S.P. thanks U. Nielsen and K. Søndergaard Larsen for useful suggestions to improve the text.

## Author contributions

T.S.P., M.O. and C.B. led the design and construction of the flux surface measurement system, based on a concept developed by M.O. and R.C.W. M.O., S.L., S.B., T.S.P., H.-S.B., R.C.W., T.K. and C.B. conducted the flux surface measurement experiments. M.O., T.S.P., S.B., S.L. and P.H. analysed the data. The article was written primarily by T.S.P., M.O. and P.H., but with input from all the co-authors. The W7-X team contributed the infrastructure, the metrology measurements and proper operation of W7-X, in particular its vacuum systems, the cryostat and the superconducting magnet system.

## Additional information

**Competing financial interests:** The authors declare no competing financial interests.

**Publisher's note**: 

## List of W7-X team members

Ivana Abramovic[5], Simppa Äkäslompolo[1], Pavel Aleynikov[1], Ksenia Aleynikova[1], Adnan Ali[1], Arturo Alonso[6], Gabor Anda[7], Tamara Andreeva[1], Enrique Ascasibar[6], Jürgen Baldzuhn[1], Martin Banduch[1], Tullio Barbui[8], Craig Beidler[1], Andree Benndorf[1], Marc Beurskens[1], Wolfgang Biel[9], Dietrich Birus[1], Boyd Blackwell[10], Emilio Blanco[6], Marko Blatzheim[1], Torsten Bluhm[1], Daniel Böckenhoff[1], Peter Bolgert[3], Matthias Borchardt[1], Lukas-Georg Böttger[1], Rudolf Brakel[1], Christian Brandt[1], Torsten Bräuer[1], Harald Braune[1], Rainer Burhenn[1], Birger Buttenschön[1], Victor Bykov[1], Ivan Calvo[6], Alvaro Cappa[6], Andre Carls[1], Bernardo Brotas de Carvalho[13], Francisco Castejon[6], Mark Cianciosa[14], Michael Cole[1,15], Stefan Costea[16], Gabor Cseh[7], Agata Czarnecka[17], Andrea Da Molin[18], Eduardo de la Cal[6], Angel de la Pena[6], Sven Degenkolbe[1], Chandra Prakash Dhard[1], Andreas Dinklage[1], Marion Dostal[9], Michael Drevlak[1], Peter Drewelow[1], Philipp Drews[9], Andrzej Dudek[1], Frederic Durodie[9], Anna Dzikowicka[19], Paul van Eeten[1], Florian Effenberg[8], Michael Endler[1], Volker Erckmann[1], Teresa Estrada[6], Nils Fahrenkamp[1], Joris Fellinger[1], Yühe Feng[1], Waldemar Figacz[17], Oliver Ford[1], Tomasz Fornal[17], Heinke Frerichs[8], Golo Fuchert[1], Manuel Garcia-Munoz[1], Benedikt Geiger[1], Joachim Geiger[1], Niels Gierse[9], Alena Gogoleva[20], Bruno Goncalves[13], Dorothea Gradic[1], Michael Grahl[1], Silvia Groß[1], Heinz Grote[1], Olaf Grulke[1], Carlos Guerard[6], Matthias Haas[1], Jeffrey Harris[14], Hans- Jürgen Hartfuß[1], Dirk Hartmann[1], Dag Hathiramani[1], Bernd Hein[1], Stefan Heinrich[1], Sophia Henneberg[1], Christine Hennig[1], Julio Hernandez[6], Carlos Hidalgo[6], Ulises Hidalgo[6], Matthias Hirsch[1], Udo Höfel[1], Hauke Hölbe[1], Alf Hölting[1], Michael Houry[21], Valentina Huber[22], Codrina Ionita[16], Ben Israeli[3], Slowomir Jablonski[17], Marcin Jakubowski[1], Anton Jansen van Vuuren[1], Hartmut Jenzsch[1], Jacek Kaczmarczyk[17], Johann-Peter Kallmeyer[1], Ute Kamionka[1], Hiroshi Kasahara[23], Naoki Kenmochi[23], Winfried Kernbichler[24], Carsten Killer[1], David Kinna[22], Ralf Kleiber[1], Jens Knauer[1], Florian Köchl[26], Gabor Kocsis[7], Yaroslav Kolesnichenko[27], Axel Könies[1], Ralf König[1], Petra Kornejew[1], Felix Köster[4], Andreas Krämer-Flecken[9], Rüdiger Krampitz[1], Natalia Krawzyk[17], Thierry Kremeyer[8], Maciej Krychowiak[1], Ireneusz Ksiazek[28], Monika Kubkowska[17], Georg Kühner[1], Taina Kurki-Suonio[29], Peter Kurz[1], Katja Küttler[1], Sehyun Kwak[1], Matt Landreman[12], Andreas Langenberg[1], Fernando Lapayese[6], Heike Laqua[1], Heinrich-Peter Laqua[1], Ralph Laube[1], Michael Laux[1], Holger Lentz[1], Marc Lewerentz[1], Yunfeng Liang[9], Shaocheng Liu[9], Jim-Felix Lobsien[1], Joaquim Loizu Cisquella[1], Daniel Lopez-Bruna[6], Jeremy Lore[14], Axel Lorenz[1],

Vadym Lutsenko[27], Henning Maaßberg[1], Jeanette Maisano-Brown[30], Oleksandr Marchuk[9], Lionello Marrelli[18], Stefan Marsen[1], Nikolai Marushchenko[1], Suguru Masuzaki[23], Kieran McCarthy[6], Paul McNeely[1], Francisco Medina[6], Dusan Milojevic[1], Alexey Mishchenko[1], Bernd Missal[1], Joseph Mittelstaedt[3], Albert Mollen[1], Victor Moncada[21], Thomas Mönnich[1], Dmitry Moseev[1], Michael Nagel[1], Dirk Naujoks[1], George Hutch Neilson[3], Olaf Neubauer[9], Ulrich Neuner[1], Tran-Thanh Ngo[21], Holger Niemann[1], Carolin Nührenberg[1], Jürgen Nührenberg[1], Marian Ochando[6], Kunihiro Ogawa[23], Jef Ongena[36], Hans Oosterbeek[1], Novimir Pablant[3], Danilo Pacella[31], Luis Pacios[6], Nerea Panadero[6], Ekkehard Pasch[1], Ignacio Pastor[6], Andrea Pavone[1], Ewa Pawelec[28], Angeles Pedrosa[6], Valeria Perseo[1], Byron Peterson[23], Dirk Pilopp[1], Fabio Pisano[32], Maria Ester Puiatti[18,31], Gabriel Plunk[1], Melanie Preynas[33], Josefine Proll[1], Aleix Puig Sitjes[1], Frank Purps[1], Michael Rack[9], Kian Rahbarnia[1], Jörg Riemann[1], Konrad Riße[1], Peter Rong[1], Joachim Rosenberger[1], Lukas Rudischhauser[1], Kerstin Rummel[1], Thomas Rummel[1], Alexey Runov[1], Norbert Rust[1], Leszek Ryc[17], Haruhiko Saitoh[1], Shinsuke Satake[23], Jörg Schacht[1], Oliver Schmitz[8], Stefan Schmuck[22], Bernd Schneider[16], Matthias Schneider[1], Wolfgang Schneider[1], Roman Schrittwieser[16], Michael Schröder[1], Timo Schröder[1], Ralf Schröder[1], Hans Werner Schumacher[34], Bernd Schweer[9], Ryosuke Seki[23], Priyanjana Sinha[1], Seppo Sipilae[29], Christoph Slaby[1], Håkan Smith[1], Jorge Sousa[13], Anett Spring[1], Brian Standley[1], Torsten Stange[1], Adrian von Stechow[1], Laurie Stephey[8], Matthew Stoneking[25], Uwe Stridde[1], Yasuhiro Suzuki[23], Jakob Svensson[1], Tamas Szabolics[7], Tamas Szepesi[7], Henning Thomsen[1], Jean-Marcel Travere[21], Peter Traverso[35], Humberto Trimino Mora[1], Hayato Tsuchiya[23], Tohru Tsuijmura[23], Yuriy Turkin[1], Swetlana Valet[1], Boudewijn van Milligen[6], Luis Vela[20], Jose-Luis Velasco[6], Maarten Vergote[36], Michel Vervier[9], Holger Viebke[1], Reinhard Vilbrandt[1], Christian Perez von Thun[22], Friedrich Wagner[1], Erhui Wang[9], Nengchao Wang[9], Felix Warmer[1], Tom Wauters[36], Lutz Wegener[1], Thomas Wegner[1], Gavin Weir[1], Jörg Wendorf[1], Uwe Wenzel[1], Andreas Werner[1], Yanling Wie[9], Burkhard Wiegel[34], Fabian Wilde[1], Thomas Windisch[1], Mario Winkler[1], Victoria Winters[8], Adelle Wright[10], Glen Wurden[11], Pavlos Xanthopoulos[1], Ichihiro Yamada[23], Ryo Yasuhara[23], Masayuki Yokoyama[23], Daihong Zhang[1], Manfred Zilker[1], Andreas Zimbal[34], Alessandro Zocco[1] & Sandor Zoletnik[7]

[5] Eindhoven University of Technology, 5612 AZ Eindhoven, Netherlands. [6] CIEMAT, Avenue Complutense, 40, 28040 Madrid, Spain. [7] Wigner Research Centre for Physics, Konkoly Thege Miklós út, H-1121 Budapest, Hungary. [8] University of Wisconsin-Madison, Engineering Drive, Madison, Wisconsin 53706, USA. [9] Forschungszentrum Jülich, Leo-Brandt-Strasse, 52428 Jülich, Germany. [10] The Australian National University, Acton ACT 2601, Canberra, Australia. [11] Los Alamos National Laboratory, P.O. Box 1663, Los Alamos, New Mexico 87545, USA. [12] University of Maryland, College Park, Paint Branch Drive, College Park, Maryland 20742, USA. [13] Instituto de Plasmas e Fusao Nuclear, Avenue Rovisco Pais 1, 1049-001 Lisboa, Portugal. [14] Oak Ridge National Laboratory, Oak Ridge, Tennessee 37831, USA. [15] Max Planck Institute for Solar System Research, Justus-von-Liebig-Weg 3, 37077 Göttingen, Germany. [16] University of Innsbruck, Innrain 52, 6020 Innsbruck, Austria. [17] Instytut Fizyki Plazmy I Laserowej Mikrosyntezy, 01 497, Hery 23, 01-497 Warszawa, Poland. [18] Istituto di Fisica del Plasma "Piero Caldirola", Via Roberto Cozzi, 53, 20125 Milano, Italy. [19] University of Szczecin, aleja Papieza Jana Pawla II 22A, Szczecin 70-453, Poland. [20] Universidad Carlos III de Madrid, Calle Madrid, 126, Getafe, Madrid 28903, Spain. [21] CEA Cadarache, 13108 St Paul lez Durance, Cedex, France. [22] Culham Science Centre, Abingdon OX14 3DB, UK. [23] National Institute for Fusion Science, 322-6 Oroshicho, Toki City, GIFU Prefecture, 509-5292, Japan. [24] Graz University of Technology, Rechbauerstrasse 12, 8010 Graz, Austria. [25] Lawrence University, 711 E Boldt Way, Appleton, Wisconsin 54911, USA. [26] Austrian Academy of Sciences, Doktor-Ignaz-Seipel-Platz 2, 1010 Wien, Austria. [27] Institute for Nuclear Research, Prospekt Nauky 47, Kyiv 03680, Ukraine. [28] Opole University, plac Kopernika 11a, Opole 45-040, Poland. [29] Department of Applied Physics, Aalto University, FI-00076 Aalto, Finland. [30] Massachusetts Institute of Technology, Cambridge, Massachusetts 02139, USA. [31] ENEA, UT Fusione, Via E. Fermi, 45, 00044 Frascati (Roma), Italy. [32] University of Cagliari, Campus Aresu—Via San Giorgio 12/2, 09124 Cagliari, Italy. [33] Ecole Polytechnique Federale de Lausanne, Route Cantonale, 1015 Lausanne, Switzerland. [34] Physikalisch-Technische Bundesanstalt, Bundesallee 100, 38116 Braunschweig, Germany. [35] Auburn University, Auburn, Alabama 36849, USA. [36] Laboratory for Plasma Physics, Ecole Royale Militaire—Koninklijke Militaire School, Avenue de la Renaissance 30, 1000 Brussels, Belgium.

DOI: 10.1038/ncomms14491    OPEN

# Erratum: Confirmation of the topology of the Wendelstein 7-X magnetic field to better than 1:100,000

T. Sunn Pedersen, M. Otte, S. Lazerson, P. Helander, S. Bozhenkov, C. Biedermann, T. Klinger, R.C. Wolf, H.-S. Bosch & The Wendelstein 7-X Team

Nature Communications 7:13493 doi: 10.1038/ncomms13493 (2016); Published 30 Nov 2016; Updated 14 Feb 2017

The original version of this Article contained errors in the spelling of authors names and their affiliations. These errors were:

The affiliation of Boyd Blackwell is 10, but was incorrectly given as 10,11,12.

In the list of the Wendelstein 7-X (W7-X) team members, the name of Matthias Borchardt was repeated twice; one instance has been removed.

In the list of the Wendelstein 7-X (W7-X) team members, the name "Shaocheng Liu" was repeated twice and, in one instance, mistakenly spelled as "Shoacheng Lui" The misspelled instance has been removed.

The name of Paul van Eeten was incorrectly spelled as Paul von Eeten.

The affiliation of Waldemar Figacz is 17, but was incorrectly given as 1.

The name of Stefan Heinrich was incorrectly spelled as Stefan Heirnich.

The affiliation of Winfried Kernbichler is 24, but was incorrectly given as 24,25.

The affiliation of Matt Landreman is 12, but was incorrectly given as 1.

The name of Henning Maaßerg was incorrectly spelled as Henning Maassberg.

The affiliation of Kieran McCarthy is 6, but was incorrectly given as 1.

The name of George Hutch Neilson was incorrectly spelled as G Hutch Neilson.

The affiliation of Jef Ongena is 36, but was incorrectly given as 9.

The affiliation of Hans Oosterbeek is 1, but was incorrectly given as 5.

The name of Maria Ester Puiatti was incorrectly spelled as Maria Ester Piulatti.

The name of Konrad Riße was incorrectly spelled as Konrad Risse.

The name of Joaquim Loizu Cisquella was incorrectly spelled as Joaquin Loizu Cisquella.

The name of Ryosuke Seki was incorrectly spelled as Ryoshuke Seki.

The affiliation of Matthew Stoneking is 25, but was incorrectly given as 1.

In the list of W7-X team members, the name of Adrian von Stechow was repeated twice; one instance has been removed.

The name of Christian Perez von Thun was incorrectly spelled as Christian Perez Von Thun.

The name of Tran-Thanh Ngo was mistakenly spelled as Tran-Tranh Ngo.

The affiliation of Adelle Wright is 10, but was incorrectly given as 10,11,12.

The affiliation of Glen Wurden is 11, but was incorrectly given as 1.

The affiliation of Masayuki Yokoyama is 23, but was incorrectly given as 1.

In affiliation 18, the Istituto di Fisica del Plasma "Piero Caldirola" was incorrectly spelled Istituto di Fisica del Plasma Piero Caldirola".

These have now been corrected in both the PDF and HTML versions of the Article.

