## [Peer Review File · Nature Communications]

Reviewer #1 (Remarks to the Author):

This paper reports first results following construction of the W7X stellarator, confirming the accuracy of the field coils by identifying the flux surfaces produced by the vacuum field. The successful construction of W7X is indeed a capstone achievement, requiring both extraordinary engineering precision and, importantly, sustained political willpower. (One need look no further than the abandoned NCSX stellarator project for a prominent contrasting example of the challenges of building such an ambitious device and the many ways such an endeavor can fail). Moreover, as suggested in the title/abstract, stellarators remain a promising path toward fusion energy—a track that may be coming of age in light of ongoing advances in high performance computing and advanced manufacturing techniques (e.g., 3D printing / additive manufacturing). Indeed, W7X will have a lot to say about the prospects of stellarators, and potentially, the viability of fusion as the ultimate energy source. In this context, the high impact of the paper is clear. The data appears to be technically sound, with substantial evidence for the conclusions. The paper is very well-written and should be accessible and interesting to a broad readership.

The weakness of the paper is that these results are preliminary, demonstrating no more than that the foundation has been laid for the start of true plasma operation—the true test of W7X as a path to fusion. The demonstration of nested flux surfaces is interesting and important, but this has also been done on earlier stellarators [Jaenicke, Nuclear Fusion 1993]. The novelty and major advances represented by this paper are the demonstration that such an ambitious and promising machine can—with unprecedented precision—be successfully built. In this context, the paper may be acceptable for publication in Nature.

Reviewer #2 (Remarks to the Author):

The paper presents the measurement of magnetic field topology in W7-X and of its deviation from the prescribed equilibrium based on the width of magnetic islands induced by externally applied resonant field errors. The reported deviations are smaller than 1 part in 100000, an outstanding engineering feat (not matched in any other magnetic fusion confinement device) that will allow the future scientific exploitation of W7-X in proving the feasibility of the stellarator concept as possible configuration for a future fusion power plant. This work is therefore of high novelty and relevance both for the general public and specialists alike and deserve publication.

The authors' approach used in supporting their claim is correct and clearly outlined. The method used to infer the island width from the images is however not described. Although the relevant references are provided, which would allow the interested reader to dive into the details, one short sentence explaining how the island width is measured (and with which uncertainty) from the Poincare plots shown would be very helpful. For example, is the quoted width w the maximum radial extent of the island? In the case shown in figure 7, the island width appears to be dominated by the size of the bright spots: is this an indication of the limit in the finite resolution with which the magnetic field topology can be measured? Addressing these two small issues would definitely strengthen the authors' conclusions.

Suggested improvements:

- 1) Lines 22 and 34: the statements that MCF is the most successful and that the stellarator is second only to the tokamak needs to be more specific. Which parameter have the authors used for this comparison? What about ICF $Q > 1$ observations? One or two additional references would be very helpful in supporting these claims.
- 2) Line 78: the authors should (briefly) explain what prompt losses are. This would help the non-specialist to fully appreciate W7-X achievements.
- 3) Lines 115 to 117 could be shortened and reference 17 is not really necessary.
- 4) The sentence in lines 121-122 does not read well. Maybe adding "providing the" after the comma?
- 5) Lines 156-157: the location of where the magnetic fields B_{mn} and B_0 are measured should be given. Also it is not clear what is meant by "overall". A brief explanation should be provided.
- 6) Line 206: is $B = 2.5$ T the same as B_0 in line 156?
- 7) Figures 4 to 7: would it be possible to add an absolutely calibrated scale of the poloidal cross-section at the toroidal location of the Poincare map?
- 8) Figure 7: the authors may want to consider adding a third panel in which the zoom in on a single island.
- 9) Figure 6: the label for the x-axis contains the poloidal flux function Ψ which is not explained in the article main text and might be somewhat obscure to the non-specialist. Maybe the authors could re-label the x-axis or give a brief explanation of the normalized poloidal flux coordinate in the figure legend.

References are appropriate and exhaustive.

The paper in all its parts is clearly written.

Point-by-point responses to the reviewers of NCOMMS-16-06882-T

Reviewer #1 (Remarks to the Author):

This paper reports first results following construction of the W7X stellarator, confirming the accuracy of the field coils by identifying the flux surfaces produced by the vacuum field. The successful construction of W7X is indeed a capstone achievement, requiring both extraordinary engineering precision and, importantly, sustained political willpower. (One need look no further than the abandoned NCSX stellarator project for a prominent contrasting example of the challenges of building such an ambitious device and the many ways such an endeavor can fail). Moreover, as suggested in the title/abstract, stellarators remain a promising path toward fusion energy—a track that may be coming of age in light of ongoing advances in high performance computing and advanced manufacturing techniques (e.g., 3D printing / additive manufacturing). Indeed, W7X will have a lot to say about the prospects of stellarators, and potentially, the viability of fusion as the ultimate energy source. In this context, the high impact of the paper is clear. The data appears to be technically sound, with substantial evidence for the conclusions. The paper is very well-written and should be accessible and interesting to a broad readership.

The weakness of the paper is that these results are preliminary, demonstrating no more than that the foundation has been laid for the start of true plasma operation—the true test of W7X as a path to fusion. The demonstration of nested flux surfaces is interesting and important, but this has also been done on earlier stellarators [Jaenicke, Nuclear Fusion 1993]. The novelty and major advances represented by this paper are the demonstration that such an ambitious and promising machine can—with unprecedented precision—be successfully built. In this context, the paper may be acceptable for publication in Nature.

We thank reviewer #1 for his or her careful reading of the manuscript, and are thankful for the positive evaluation that the paper fits Nature Communications. We have further added one more new result - that the 'as-built, as installed' geometry of the coils based on metrology accurately predicts the phase and size of the measured intrinsic error field, strengthening our confidence in our engineering as well as in the measurement method.

Reviewer #2 (Remarks to the Author):

The paper presents the measurement of magnetic field topology in W7-X and of its deviation from the prescribed equilibrium based on the width of magnetic islands induced by externally applied resonant field errors. The reported deviations are smaller than 1 part in 100000, an outstanding engineering feat (not

matched in any other magnetic fusion confinement device) that will allow the future scientific exploitation of W7-X in proving the feasibility of the stellarator concept as possible configuration for a future fusion power plant. This work is therefore of high novelty and relevance both for the general public and specialists alike and deserve publication.

The authors' approach used in supporting their claim is correct and clearly outlined. The method used to infer the island width from the images is however not described. Although the relevant references are provided, which would allow the interested reader to dive into the details, one short sentence explaining how the island width is measured (and with which uncertainty) from the Poincare plots shown would be very helpful. For example, is the quoted width w the maximum radial extent of the island? In the case shown in figure 7, the island width appears to be dominated by the size of the bright spots: is this an indication of the limit in the finite resolution with which the magnetic field topology can be measured? Addressing these two small issues would definitely strengthen the authors' conclusions.

We thank the reviewer for this comment. We have added a section that describes how the island width is determined and with what accuracy. We reproduce the added text below for convenience:

Light fibers installed in the vessel along with detailed measurements of their location allow the pixels of the image plane to be mapped to physical dimensions. In this way the width of the island in physical units can be inferred from a measurement in pixels. Error bars account for both the physical width of the flux surface traces and the step size going from outside the island chain to inside it. A best attempt is made to report the maximum width of the magnetic islands.

Suggested improvements:

- 1) Lines 22 and 34: the statements that MCF is the most successful and that the stellarator is second only to the tokamak needs to be more specific. Which parameter have the authors used for this comparison? What about ICF $Q > 1$ observations? One or two additional references would be very helpful in supporting these claims.

Following the editors' recommendation, we have reduced the introductory text significantly. Following the suggestion above would make the introduction longer again. We have instead reformulated the sentence to address what we see as the reviewer's primary concern: that our claim was not sufficiently substantiated. We made a less sweeping statement that we hope is uncontroversial and does not need further elaboration.

- 2) Line 78: the authors should (briefly) explain what prompt losses are. This would help the non-specialist to fully appreciate W7-X achievements.

We have replaced the "prompt orbit losses" phrase with the previously introduced

concept of unconfined drift-orbits. The two are identical.

3) Lines 115 to 117 could be shortened and reference 17 is not really necessary.

Done: The sentence has been shortened and the reference removed

4) The sentence in lines 121-122 does not read well. Maybe adding "providing the" after the comma?

We agree and have reformulated the sentence

5) Lines 156-157: the location of where the magnetic fields B_{mn} and B_0 are measured should be given. Also it is not clear what is meant by "overall". A brief explanation should be provided.

B_{mn} and B_0 are not local quantities; they are Fourier coefficient amplitudes. This is explained in lines 155-156. We reformulated the sentence to make it clearer, also to address the reviewer's point 6) below.

6) Line 206: is $B = 2.5$ T the same as B_0 in line 156?

No, but see the answer to 5 and the revisions in the manuscript.

7) Figures 4 to 7: would it be possible to add a an absolutely calibrated scale of the poloidal cross-section at the toroidal location of the Poincare map?

This is a very good suggestion – we revised Figures 4 and 7 accordingly.

8) Figure 7: the authors may want to consider adding a third panel in which the zoom in on a single island.

Due to the shadowing issues, we are currently unable to measure any internal structure of the island chains. A zoom-in of the island will therefore not reveal anything extra. We have in the meanwhile built in a re-designed electron gun that should allow us to see more details inside the islands in measurements to be performed in 2017.

9) Figure 6: the label for the x-axis contains the poloidal flux function Ψ which is not explained in the article main text and might be somewhat obscure to the non-specialist. Maybe the authors could re-label the x-axis or give a brief explanation of the normalized poloidal flux coordinate in the figure legend.

We have added text in the Figure caption to explain the meaning of the x-axis

References are appropriate and exhaustive.

The paper in all its parts is clearly written.

In addition to the above-mentioned changes, we have added a new result showing that the measured field error is consistent with that calculated from a numerical model that includes the measured coil deviations.

We have also added a short note explaining what the guiding center is.

Reviewer #2 (Remarks to the Author):

Only a few minor comments on the revised paper:

1) figure 8: maybe it is not necessary to show the linear fit, unless an explanation for the offset for zero trim current is clearly stated. Also, why have the theoretical data point slightly higher trim current?

2) lines 226-227: not entirely clear how the agreement concerning the phase of the island can be inferred from figure 8: maybe a clarification would help.

3) In their rebuttal, the authors reproduce the text explaining how the island width is determined which they claimed it was added to the revised manuscript, however the revised manuscript does not contain it.

Responses to reviewer comments

Reviewer comments in italics, our responses in blue:

Only a few minor comments on the revised paper:

1) figure 8: maybe it is not necessary to show the linear fit, unless an explanation for the offset for zero trim current is clearly stated.

We now explicitly describe the meaning of the different lines in the legend.

Also, why have the theoretical data point slightly higher trim current?

The numerical points were chosen to not coincide exactly with the experimental ones in order for the reader to more easily distinguish the symbols. We made no changes to the manuscript in this regard.

2) lines 226-227: not entirely clear how the agreement concerning the phase of the island can be inferred from figure 8: maybe a clarification would help.

We have clarified in the text that Figure 8 only shows the agreement regarding amplitude.

3) In their rebuttal, the authors reproduce the text explaining how the island width is determined which they claimed it was added to the revised manuscript, however the revised manuscript does not contain it.

We thank the reviewer for having caught this mistake. The text is now included.